Genomic characterization and phylogenetic analysis of the first SARS-CoV-2 variants introduced in Lebanon

Feghali Rita 1
http://orcid.org/0000-0002-5558-4088 Merhi Georgi 2
http://orcid.org/0000-0003-0826-613X Kwasiborski Aurelia 3
Hourdel Veronique 3
Ghosn Nada 4
Tokajian Sima 2 stokajian@lau.edu.lb
1 Department of Laboratory Medicine, Rafik Hariri University Hospital , Beirut , Lebanon
2 Department of Natural Sciences, Lebanese American University , Byblos , Lebanon
3 Laboratory: Environment and Infectious Risks, Pasteur Institute , Paris , France
4 Epidemiological Surveillance Unit, Ministry of Public Health , Beirut , Lebanon
Bolshoy Alexander
Electronic publication date: 2021 Mar 16
Publication date: 2021
Volume: 9
Electronic Location ID: e11015
Received 2020 Sep 23; Accepted 2021 Feb 5
Copyright: © 2021 Feghali et al.
Copyright year: 2021
Copyright holder: Feghali et al.
License: This is an open access article distributed under the terms of the Creative Commons Attribution License, which permits unrestricted use, distribution, reproduction and adaptation in any medium and for any purpose provided that it is properly attributed. For attribution, the original author(s), title, publication source (PeerJ) and either DOI or URL of the article must be cited.
License URL: https://creativecommons.org/licenses/by/4.0/

Keywords: COVID-19, SASR-CoV-2, Lebanon, SNV analysis, B.1 (20A clade), B.4 (19A clade)

Funding: MediLabSecure SARS-CoV-2 European Commission (DEVCO) IFS/2018/402-247 Strategic Research Review Committee #SRRC-R-2019-38 Lebanese American University and by the National Council for Scientific Research #00993 The sequencing of the SARS-CoV-2 samples was financed by the MediLabSecure project, founded by the European Commission (DEVCO: IFS/2018/402-247). This study was also funded by the Strategic Research Review Committee (Grant #SRRC-R-2019-38) at the Lebanese American University and by the National Council for Scientific Research (Grant #00993). The funders had no role in study design, data collection and analysis, decision to publish, or preparation of the manuscript.

==============================
Background

In December 2019, the COVID-19 pandemic initially erupted from a cluster of pneumonia cases of unknown origin in the city of Wuhan, China. Presently, it has almost reached 94 million cases worldwide. Lebanon on the brink of economic collapse and its healthcare system thrown into turmoil, has previously managed to cope with the initial SARS-CoV-2 wave. In this study, we sequenced 11 viral genomes from positive cases isolated between 2 February 2020 and 15 March 2020.

Methods

Sequencing data was quality controlled, consensus sequences generated, and a maximum-likelihood tree was generated with IQTREE v2. Genetic lineages were assigned with Pangolin v1.1.14 and single nucleotide variants (SNVs) were called from read files and manually curated from consensus sequence alignment through JalView v2.11 and the genomic mutational interference with molecular diagnostic tools was assessed with the CoV-GLUE pipeline. Phylogenetic analysis of whole genome sequences confirmed a multiple introduction scenario due to international travel.

Results

Three major lineages were identified to be circulating in Lebanon in the studied period. The B.1 (20A clade) was the most prominent, followed by the B.4 lineage (19A clade) and the B.1.1 lineage (20B clade). SNV analysis showed 15 novel mutations from which only one was observed in the spike region.

Introduction

In December 2019, unknown cases of pneumonia were detected in the city of Wuhan, Hubei province, China. Infected individuals exhibited symptoms similar to that of severe acute respiratory syndrome (SARS) (Li et al., 2020). Deep sequencing identified the causative agent as a novel β-coronavirus, named nCoV-2019, later, renamed as SARS-CoV-2 (Gralinski & Menachery, 2020). Since then, the World Health Organization (WHO) has classified the spread of the virus as a global pandemic (Astuti & Ysrafil, 2020) and as of 18 January 2021, there was a total of 93,194,922 confirmed cases with 2,014,729 deaths (https://covid19.who.int/). COVID-19’s common clinical symptoms include, but not limited to, fever, development of a dry cough, myalgia/fatigue, dyspnea, headaches and pneumonia (Zhou et al., 2020; Huang et al., 2020).

Molecular characterization of the SARS-CoV-2 genome showed 79.6% sequence similarity to SARS-CoV and 96% to the RaTG13 bat-CoV (Zhou et al., 2020) supporting that bats (Rhinolophus affinis) may have acted as a reservoir for the SARS-CoV-2 progenitor (Andersen et al., 2020). Interestingly, analysis of SARS-CoV-2’s structural protein ORF3a revealed highly conserved protein domains within homologs from civet, pangolin and SARS-like bat-CoVs (Issa et al., 2020).

Despite the difficult economic situation and the waning infrastructure of its healthcare sector, Lebanon has successfully managed the initial COVID-19 wave. As of 21 February 2020, there has been 249,158 confirmed cases with 1,866 total deaths, with 1.4% of cases being attributed to exposure outside Lebanon and 98.6% due to local community spread of the SARS-CoV-2 virus (https://www.moph.gov.lb/en/Pages/2/193/esu#/en/Pages/2/24870/novel-coronavirus-2019-).

In this study, we performed a comprehensive genomic analysis of 11 SARS-CoV-2 isolates recovered from Lebanese individuals during the first phase of the pandemic. We also looked at their phylogeny based on location and date of exposure. Finally, we determined the single nucleotide variants (SNVs) and amino acid changes and compared our results with worldwide disseminating SARS-CoV-2 variants to assess epidemiological relatedness.

Materials and Methods

COVID-19 response in Lebanon and genomic sequencing

The 11 cases (Table 1) undertaken in this study were detected and then selected randomly between 21 February 2020 and 15 March 2020 and designated as S1–S11. The clinical data were collected as part of the quarantine monitoring measures at the Rafik al Hariri University Hospital with the support of the Lebanese Ministry of Public Health. Required written informed consent was not obtained for the first studied cases due to the pressing need for data collection in the early stages of the outbreak. This study was approved by the Institutional Review Board (IRB) of the Lebanese American University (IRB #:AU.SAS.ST2.19/May/2020).

Table 1 Clinical characteristics of Lebanese patients with COVID-19 between February and March 2020.

Accession number—GISAID	Sample	Gender	Age (years)	Location of exposure	Sample type	Ct*	Patient status	Signs and symptoms	
EPI_ISL_450508	S1	Male	56	Egypt	Sputum/PBS	14.78	Hospitalized-deceased	Early stages: flu like symptoms
Followed by severe dyspnea and severe ARDS†
Chest X-ray: patchy bilateral upper lobe consolidation	
EPI_ISL_450509	S2	Male	63	Local—community acquired	Nasopharyngeal VTM	26.15	Hospitalized-released	Asymptomatic	
EPI_ISL_450510	S3	Male	19	Iran	Nasopharyngeal VTM	37.5	Hospitalized-released	Headache, abdominal pain and diarrhea	
EPI_ISL_450511	S4	Female	33	United Kingdom	Nasopharyngeal VTM	34.67	Hospitalized-released	Rhinorrhea and headache	
EPI_ISL_450512	S5	Male	42	Iran	Nasopharyngeal VTM	16.11	Hospitalized-released	Asymptomatic	
EPI_ISL_450513	S6	Male	–	France	Nasopharyngeal VTM	33.7	Hospitalized-released	Asymptomatic	
EPI_ISL_454420	S7	Female	41	Iran	Nasopharyngeal VTM	20.24	Hospitalized-released	Sore throat and rhinorrhea	
EPI_ISL_450514	S8	Female	74	Local—community acquired	Nasopharyngeal VTM	33.8	Hospitalized-released	Dyspnea	
EPI_ISL_450515	S9	Female	25	United Kingdom	Nasopharyngeal VTM	34	Hospitalized-released	Asymptomatic	
EPI_ISL_450516	S10	Male	36	Local—community acquired	Sputum/PBS	–	Hospitalized-released	Asymptomatic	
EPI_ISL_450517	S11	Male	22	Italy	Nasopharyngeal VTM	–	Hospitalized-released	Early stages: dysuria, fever and flu-like symptoms
Followed by severe dyspnea and ARDS
Chest CT: bilateral infiltrates with ground glass appearance	
Notes:

* CT, RT-PCR Cycle Threshold.

† ARDS, acute respiratory distress syndrome.

RNA was extracted from the specimens using the Qiagen QIAamp Viral RNA Mini kit (QIAGEN, Hilden, Germany) by following the manufacturer’s instructions. An qRT-PCR corresponding to the Charité protocol (published on 17 January 2020) was used for detection of SARS-CoV-2. The assay relies on a first-line E gene screening, followed by a confirmatory assay using the RNA dependent RNA polymerase (RdRp) gene and a synthetic RNA positive control (Charité virology institute—Universitätsmedizin Berlin, Berlin, Germany). A 25 µl reaction was set up containing 1 µl of forward primer (10 µM), 1 µl of reverse primer (10 µM), 0.5 µl of probe (10 µM), 5 µl (100 ng/µl) of extracted RNA, 12.5 µl of 2X reaction buffer and 1 µl of reverse Transcriptase/Taq Polymerase mixture (Invitrogen, New York, NY, USA) provided with the Superscript III One step RT-PCR system (Corman et al., 2020).

Thermal cycling was performed at 55 °C for 10 min for reverse transcription, followed by 95 °C for 3 min and then 45 cycles of 95 °C for 15 s and 58 °C for 30 s. cDNA amplification was done through the tiled amplification approach and following ARCTIC’s network recommended protocol (Quick, 2020).

Sequencing libraries were prepared from tiled amplicons using the Miseq reagent kit v3 (Illumina, San Diego, CA, USA) and sequenced on an Illumina MiSeq system using 250 bp paired-end reads and following the manufacturer’s instructions. Sequences were randomly labeled from S1 to S11 based on their loading order onto the sequencer.

Consensus sequences

Raw sequencing reads were input into FastQC v0.11.9 (https://www.bioinformatics.babraham.ac.uk/projects/fastqc/) for quality assessment and subsequent quality control was performed through FQCleaner v3.0 (Criscuolo & Brisse, 2013) with a 28 quality score threshold and a minimum read length of 30. Quantified reads were mapped to the Wuhan-Hu-1 reference genome (MN908947.3) using BWA v0.7.17 with the “MEM” alignment algorithm (https://arxiv.org/abs/1303.3997). Consensus sequences were generated for all 11 SARS-CoV-2 isolates using SAMtools v1.9 (http://www.htslib.org/) and vcftools v0.1.16 (https://vcftools.github.io/index.html).

Phylogenetic analysis

Six high quality (N < 0.5%; Length > 29,000 bp) genomes (S1, S2, S4, S5, S8, S9) were selected for phylogenetic analysis. Reference sequences (n = 55) were chosen based on several criteria including phylogenetic placement of reference genomes, collection dates, history of exposure (travel), GISAID’s BLAST feature within the EpiCoV™ browser (https://www.gisaid.org) and overall genome quality/completeness to avoid sequence-based bias (Table S1).

Sequences were aligned with MAFFT v7.467 (Katoh & Standley, 2013). The resulting alignment was used in masking terminal regions and gaps with Nextstrain’s custom Python script (https://github.com/nextstrain/ncov/tree/master/scripts). The alignment was input into ModelFinder to assess the best fit substitution model (Kalyaanamoorthy et al., 2017). A maximum likelihood (ML) tree was generated with IQ-TREE v2 (Minh et al., 2020) using the TIM2+F substitution model. Boot-strap support was established through 1,000 iterations for Ultra-Fast Bootstrapping (UFBoot) and SH-like approximate likelihood ratio test (SH-aLRT) (Guindon et al., 2010). The consensus tree was visualized with the interactive tree of life v4 (IToL; https://itol.embl.de/) (Letunic & Bork, 2019).

We assigned genetic lineages based on three commonly used systems including the recently proposed dynamic classification using Pangolin tool v1.1.14 (https://github.com/hCoV-2019/pangolin) (Rambaut et al., 2020), the Nextstrain classification and GISAID’s internal classification.

Comparative genome and spike (S) protein analyses

SAM files from read alignment for all samples were converted into BAM files with SAMtools v1.9 (http://www.htslib.org/). Using bcftools v1.9 (https://samtools.github.io/bcftools/), variants were called and extracted through the “mpileup” and “call” commands with ploidy set to 1 and invoking the multiallelic-caller through the “-m” flag. Obtained variants were filtered with the “varfilter” command using the custom perl script vcfutils.pl (http://www.htslib.org/).

Consensus genomes were aligned against Wuhan-Hu-1 (MN908947.3) with MAFFT v7.467 (Katoh & Standley, 2013) and the resulting alignment was input in SNP-sites v2.5.1(Page et al., 2016) to extract all identified SNPs (Table 2). The alignment was visualized (Fig. S1) with JalView v2.11 (Waterhouse et al., 2009) and polymorphic sites were manually curated (positions, with low coverage, defined by N strings were omitted). For added stringency, sequences were input into the CoV-GLUE analysis pipeline (http://cov-glue.cvr.gla.ac.uk/) where all SNPs and amino acid variations were identified for all genomes. Potential interference with all available diagnostic assays for SARS-CoV-2 was also investigated.

Table 2 SNP distribution in SARS-CoV-2 genome sequences.

Gene	High depth (DP > 15) nucleotide mutations	Low depth (DP < 15) nucleotide mutations	Isolates	
Nsp2	G1397A	C884T, C1093T, C1884T	S5, S9	
Nsp3	C3037T, A7766G	C6078T, C6198A, A6281G, A6282T, C6285A, C7528T,	S1, S2, S4, S5, S7, S8, S10, S11	
Nsp4	C9118T	G8653T, C8655T, A8658G, A8897T, T9860C, T9861G	S2, S5, S7, S8, S9, S10, S11	
Nsp5	–	C10074T, A10075T	S7, S11	
Nsp6	G11083T	–	S5, S9	
Nsp8	–	A12297T	S10	
Nsp10	C13381T	–	S2, S8, S11	
Nsp12	C14408T	G14369T, C14703T, C14724T, C14802T, C14993T	S1, S2, S4, S7, S8, S10, S11	
Nsp13	–	G16301T	S7	
Nsp14	C18877T	G18670T	S1, S7	
Nsp15	–	A19499C	S7	
Spike (S) gene	A23403G	G22021T, T22092G	S1, S2, S4, S7, S8, S9, S10, S11	
ORF3a	G25563T, C25578T, C25609T	C25611T, A25965G	S1, S2, S5, S8, S9, S10, S11	
ORF7a	–	C27643T	S11	
None-coding region	–	G27788A, T27789C	S5, S9	
N gene	T28688C, G28881A, G28882A, G28883C	C28354T	S4, S5, S8, S9	
None-coding region	–	G29543T	S7	
3′ UTR	G29742T	–	S5, S7, S9	

Genome annotation was performed with Prokka v1.14.6 (Seemann, 2014). Subsequently, spike (S) protein amino acid sequences were extracted, aligned with MAFFT v.7467 (Katoh & Standley, 2013) and visualized with JalView v2.11 (Waterhouse et al., 2009).

Results

Clinical characteristics of patients

The first SARS-CoV-2 positive case was documented in Lebanon on 21 February 2020. By 15 March 2020, Lebanon had a total of 108 positive cases (https://www.moph.gov.lb/maps/covid19.php). Among the eleven patients, five were clinically asymptomatic (n = 5) and four exhibited mild symptoms (n = 4). The remaining two patients displayed a severe form of COVID-19 (Fig. 1). All patients were hospitalized, including asymptomatic carriers as a form of quarantined isolation to slow community spread (Fig. 1). Four patients out of eleven (n = 4) were female and the median age was 38.5 years and the range was 19–74 (Table 1).

Figure 1 Timeline of symptom onset, illness severity, SARS-CoV-2 RNA sample collection date, hospitalization, and ICU admission of the 11 COVID-19 patients between February and March 2020 in Lebanon.

All patients were admitted to the hospital with varying illness severity. Light blue bars represent asymptomatic patients. Gold bars indicate patients displaying mild symptoms while red bars denote patients with severe cases. Days are numbered sequentially from the 17th of February until the 20th of March. The star symbol indicates the RNA sample collection date while the asterisk (*) symbol denotes the date of symptom onset. Orange colored triangles mark the initial hospitalization date. The target symbol indicates admission into the intensive care unit (ICU) and the plus (+) sign is unique to patient S11 where it indicates his admission into the ICU in a different healthcare facility. The forked arrow symbol represents a patient’s death.

Dates of symptom onset varied between the 17th of February and 13th of March (Fig. 1). Travel history differed between patients and included countries such as Iran, France, Italy and the United Kingdom (Table 1), which was consistent with multiple introduction incidences. The initial signs and symptoms were headache (n = 2), rhinorrhea (n = 2) and flu-like symptoms (n = 2). Over the course of illness, a patient reported abdominal pain and diarrhea while another only suffered from dyspnea (Table 1). The two patients with severe COVID-19, initially developed flu-like symptoms followed by severe dyspnea and acute respiratory distress syndrome (Fig. 1).

Phylogenetic analysis

The SARS-CoV-2 isolates recovered from Lebanon clustered, and according to Nextstrain’s classification, in three distinct clades namely: 19A (93% bootstrap support), 20A (81% bootstrap support) and 20B (98% bootstrap support) (Fig. 2). S5 (EPI_ISL_450512) and S9 (EPI_ISL_450515) were grouped closely to sequences from India (EPI_ISL_435106, EPI_ISL_421667, EPI_ISL_435101) and Kuwait (EPI_ISL_416458) in clade19A. Interestingly, S9 displayed less phylogenetic divergence than S5 based on the distance observed in the phylogenetic tree and polymorphic sites differences. Both patients had different exposure histories (Table 1).

Figure 2 A maximum-likelihood (ML) phylogenetic tree of SARS-CoV-2 genomic sequences isolated from Lebanon.

Only six genomes, with labels colored red, were selected for the phylogenetic tree to prevent any bias due to N strings in genomic sequences. Bootstrap values are represented by dark blue circles on the tree branches. Nextstrain clades were defined through different clade colors with light blue denoting the 19A clade, dark blue representing the 20B clade and gold designating the 20A clade. Pangolin lineages are displayed by star symbols. Purple stars represent the B.4 lineage, brown denotes the B.1.1 lineage and orange-colored stars stand for the B.1 lineage. Tree scale connotes for raw branch length and an internal scale system was added for additional stringency.

S1 (EPI_ISL_450508), S2 (EPI_ISL_450509) and S8 (EPI_ISL_450514) clustered within clade 20A. S2 and S8 were closely related to one recovered from Egypt (EPI_ISL_430820). However, S2 was phylogenetically more related to the isolate from Egypt than S8, with both being linked to local community transfer (Fig. 2; Table 1). S1 was recovered from a patient with travel history to Egypt and clustered close to sequences recovered from Saudi Arabia (EPI_ISL_437697), USA (EPI_ISL_447844) and Taiwan (EPI_ISL_444276) (Table 1). Additionally, S1 showed lesser evolutionary distance than S2 and S8 (Fig. 2) indicating less genomic diversity. S4 (EPI_ISL_450511) on the other hand, was of the same superclade as that of S1, S2 and S8, but clustered under 20B (Fig. 2). It also showed proximity to sequences recovered from Europe and more so from England (EPI_ISL_448804), Belgium (EPI_ISL_417422) and Lithuania (EPI_ISL_450496), which was in accordance with the patient’s travel history.

We also assigned the lineages using pangolin v1.1.14, and the results (Table 3) obtained were consistent (Fig. 2). S5 and S9 were assigned to the B.4 lineage alongside S3 (Table 3). S4 and S6 were assigned to lineage B.1.1, while the remaining isolates clustered under the B.1 lineage representing clade 20A (Fig. 2). The GISAID nomenclature could be also correlated with the lineage assignments. S5 and S9 were included in the O clade with all the others fitting under the super G clade: S4 and S6: GR clade and S1, S2, S7, S8, S10 and S11: GH clade.

Table 3 Novel amino acid mutations in SARS-CoV-2 genomes.

Sample	Total number of SNVs	Genome coverage (%)	Novel amino acid changes as of time of submission	Pangolin lineage assignment	SH-alrt (%)	UFbootstrap (%)	
S1	5	>95	None	B.1	100	100	
S2	6	>95	None	B.1	100	100	
S3	n/a	<95	n/a	B.4	100	100	
S4	6	>95	None	B.1.1	100	99	
S5	8	>95	nsp3: I1683V	B.4	100	100	
S6	n/a	<95	n/a	B.1	100	100	
S7	16	>95	nsp12: C310F; nsp13: R22I; nsp14: D211Y, H487P	B.1	100	100	
S8	10	>95	nsp3: S1160Y; nsp4: M33Ffs	B.1	100	100	
S9	7	>95	None	B.4	100	100	
S10	18	>95	nsp3: N1188V, T1189N; nsp4: L436R; nsp8: Q69L; ORF7b: C12R	B.1	100	100	
S11	19	>95	nsp4: N115Y; nsp12: S518L; S protein: M177R	B.1	100	100	

SNVs in sequenced genomes and the S protein

The obtained consensus genomes of S3 and S6 showed low coverage values being 70.2% and 92.6%, respectively. Accordingly, polymorphism could not be confirmed and as such were excluded from downstream analysis. Table 2 shows all detected SNP (high and low depth) sites across the 11 aligned genomes. The genetic variation within the aligned genomes was relatively low, with a minimum of 5 and a maximum of 19 SNPs (median: 11 SNPs). In S1, S2, S7, S8, S10 and S11 three amino acid (AA) changes were detected within: ORF1ab within the non-structural protein 12 (nsp12) P323L, spike (S) protein D614G and ORF3a’s Q57H (Table 2). Only P323L and D614G AA changes were detected in S4 with two other additional mutations in the N protein (R203K and G204R). S5 and S9 shared four common AA changes in nsp2 (R27C, V198I), nsp4 (M33I) and nsp6 (L37F). Interestingly, S5 had an additional mutation in nsp2 (A360V) and a novel mutation in nps3 (I1683V), while S9 displayed an AA change in the spike (S) protein at position 153 (M153I).

We also detected in S7, S8, S10 and S11 novel amino acid changes in multiple coding regions within the SARS-CoV-2 genome (Table 3). Furthermore, S8 had a novel frame-shift (fs) deletion at nucleotide position 8,651, leading to a change in nsp4 at position 33 and as a result replacing a methionine residue by the aromatic residue phenylalanine (M33Ffs).

Analysis of the polymorphic sites in the context of diagnostics revealed multiple hot-spots where accumulating nucleotide polymorphisms could interfere with the diagnostic schemes based on the ARCTIC network amplicon sequencing primers. In particular, changes such as: C13381T, G28881A, G28882A and G28883C in S2, S4, S6, S8 and S11 may interfere with the specificity of the primers and probes designed by the Chinese Center for Disease Control and Prevention (China CDC) and used for the detection of 2019-nCoV through targeting the ORF1ab encoded polymerase and N protein.

We also compared the S protein within the 11 genomes through an intra-isolate alignment. The D614G amino acid change was detected in all the isolates except S3, S5 and S9. This mutation, however, would not cause changes in the receptor binding domain (RBD-Spike: 455–505) or in the polybasic cleavage site unique (PCS-Spike: 681–686) to SARS-CoV-2. Finally, M153I and M177R mutations were only observed in isolates S9 and S11, respectively.

Discussion

We aimed in this study at studying the SARS-CoV-2 isolates recovered at the early stages of the COVID-19 outbreak in Lebanon. To that end, we sequenced eleven genomes, including patient zero, collected from 21 February 2020 to 15 March 2020. We investigated the phylogenetic and epidemiological relatedness of the genomes and found three lineages circulating since the start of the local outbreak. Furthermore, several novel amino acid mutations in ORF1ab encoding for non-structural proteins and other structural proteins were detected.

Our results showed that S3 and S5 belonged to the B.4 lineage (O clade on GISAID) with a travel history linked to Iran (Table 1). This is consistent with previous reports where distinct clades were attributed to returnees from Iran, with all the genomes clustering under the B.4 lineage (O clade) (Eden et al., 2020; Potdar et al., 2020). Moreover, sequences that clustered close to S5 in Fig. 2 were also associated with possible exposure and travel history to Iran. It is noteworthy, that the first COVID-19 case in Iran was officially reported on 19 February (https://covid19.who.int/region/emro/country/ir) whereas the first case (S7) in Lebanon was on 21 February and linked to a patient returning from the city of Qom, Iran (https://www.moph.gov.lb/en/Media/view/27426/coronavirus-disease-health-strategic-preparedness-and-response-plan-), and which clustered under the B.1 lineage (GH clade).

S4 and S9 were however, both linked to patients with travel history to the United Kingdom (UK), but clustered in two different lineages, B.1.1 and B.4, respectively. Detailed analysis of the European sub-clusters by Nextstrain (Hadfield et al., 2018) revealed that the UK outbreak was largely rooted in B.1 and B.1.1 lineages which was in agreement with the analysis from GISAID’s EpiCoV™ database (Shu & McCauley, 2017). The outbreak in Europe was mainly linked to isolates clustering under clade G and its variants GH/GR. Available data suggested that sequences linked to travel history to the UK represented the early stages of the outbreak with the B.4 lineage being actively circulating at the time.

The predominant lineage in this study was B.1 (clade GH) (S1, S2, S7, S8, S10, S11). Figure 2 shows that S2 and S8 were closely related to SARS-CoV-2 recovered in Egypt (EPI_ISL_430820) and somewhat distant from S1. S1 was linked to travel history to Egypt, while S2 and S8 to local community spread suggesting potential multiple introductions especially with the observed discrepancies in the number of SNPs in S8 compared to S1 (Table 3).

S protein amino acid changes revealed three variants among the sequenced isolates. Previously, Bhattacharyya et al. (2020) suggested the widespread dominance of SARS-CoV-2 with D614G/A23403G substitution in Europe and North America (Table 2). The delC allele in TMPRSS2, common in Europe and North America, facilitated the entry of the 614G subtype into host cells, thus accelerating the spread of 614G subtype (Bhattacharyya et al., 2020).

A novel S protein mutation (M177R) was detected in the S1 subunit in one of the isolates in this study (S11) (Table 3), but at this point it is not clear whether it has any implications on its interaction with the ACE2 receptor. Additionally, the RdRp mutation P323L (C14408T), generally detected in isolates recovered from Europe and North America (Pachetti et al., 2020), was also detected in the sequenced isolates from Lebanon, and was consistent with the metadata further showing multiple introduction points from Europe (UK, Italy, France). Mutations in the RdRp are of interest with the encoded polymerase being an important target for current therapeutic polymerase inhibitors (Pachetti et al., 2020), and with our data revealing several novel mutations in nsp12 (Table 3).

Conclusions

The estimated mutation rate driving the genomic global diversity of the virus has been determined at approximately 6 × 10−4 nucleotides/genome/year (Van Dorp et al., 2020). The accumulated genetic diversity in the form of SNVs in the SARS-CoV-2 genomes throughout the COVID-19 pandemic serves as a tool to assess and quantify the genomic diversity, evolutionary distribution, and epidemiological linkage of the virus (Yang et al., 2020). With daily confirmed cases on an exponential rise in Lebanon (https://www.moph.gov.lb/en/Pages/2/193/esu#/en/Pages/2/24870/novel-coronavirus-2019-) and the rapid emergence of novel variants of concern (Volz et al., 2021), further sequencing efforts are urgently needed to assess the spread and phylogenomic characteristics of SARS-CoV-2 in Lebanon and keep track of the emerging variants which is much needed to mitigate the spread, and for vaccine development and efficacy. This study offers a comprehensive genomic snapshot of the earlier stages of the local outbreak. Importantly, our analysis highlights the viral lineages and genomic mutations identified at the root of the Lebanese outbreak which are also reflective of the situation in neighboring regions that lack genomic and epidemiological data.

Supplemental Information

Supplemental Information 1 Visualization of the 11 SARS-CoV-2 alignment against the Wuhan-Hu-1 (MN908947.3) reference genomes and observed single nucleotide variants (SNVs) using JalView v2.11.1.3.

Click here for additional data file.

Supplemental Information 2 Metadata for all (n = 55) SARS-CoV-2 reference genome downloaded from GISAID’s EpiCoV database.

Metadata table includes isolate name, accession number, collection date, location, travel history, and lineage information.

Click here for additional data file.

Supplemental Information 3 Consensus sequence for SARS-CoV-2 sample S1 with 99.7% coverage.

Click here for additional data file.

Supplemental Information 4 Consensus sequence for SARS-CoV-2 sample S2 with 99.7% coverage.

Click here for additional data file.

Supplemental Information 5 Consensus sequence for SARS-CoV-2 sample S3 with 70.2% coverage.

Click here for additional data file.

Supplemental Information 6 Consensus sequence for SARS-CoV-2 sample S6 with 92.6% coverage.

Click here for additional data file.

Supplemental Information 7 Consensus sequence for SARS-CoV-2 sample S5 with 99.8% coverage.

Click here for additional data file.

Supplemental Information 8 Consensus sequence for SARS-CoV-2 sample S10 with 97.1% coverage.

Click here for additional data file.

Supplemental Information 9 Consensus sequence for SARS-CoV-2 sample S7 with 99.7% coverage.

Click here for additional data file.

Supplemental Information 10 Consensus sequence for SARS-CoV-2 sample S4 with 99.7% coverage.

Click here for additional data file.

Supplemental Information 11 Consensus sequence for SARS-CoV-2 sample S11 with 98.2% coverage.

Click here for additional data file.

Supplemental Information 12 Consensus sequence for SARS-CoV-2 sample S8 with 99.7% coverage.

Click here for additional data file.

Supplemental Information 13 Consensus sequence for SARS-CoV-2 sample S9 with 99.7% coverage.

Click here for additional data file.

We thankfully acknowledge the personnel and laboratories who have generated and submitted sequences to the GISAID’s EpiCoV™ database. This study does not declare ownership of these sequences. The openly available data was used to compare our results within an international framework and to provide further information about the phylogenomic status and the spreadability of the SARS-CoV-2 in countries with little to non-existing epidemiological and genomic data. We also acknowledge Dr. Guillain Mikaty, Dr. Valerie Caro and Dr. Jean-Claude Manuguerra from Institut Pasteur, for their technical assistance and support in the sequencing of the SARS-CoV-2 samples.

Additional Information and Declarations

Competing Interests

Author Contributions

Human Ethics

DNA Deposition

Data Availability

The authors declare that they have no competing interests.

Rita Feghali conceived and designed the experiments, performed the experiments, analyzed the data, prepared figures and/or tables, authored or reviewed drafts of the paper, and approved the final draft.

Georgi Merhi conceived and designed the experiments, performed the experiments, analyzed the data, prepared figures and/or tables, authored or reviewed drafts of the paper, and approved the final draft.

Aurelia Kwasiborski conceived and designed the experiments, performed the experiments, authored or reviewed drafts of the paper, and approved the final draft.

Veronique Hourdel conceived and designed the experiments, performed the experiments, authored or reviewed drafts of the paper, and approved the final draft.

Nada Ghosn conceived and designed the experiments, performed the experiments, authored or reviewed drafts of the paper, and approved the final draft.

Sima Tokajian conceived and designed the experiments, performed the experiments, analyzed the data, prepared figures and/or tables, authored or reviewed drafts of the paper, and approved the final draft.

The following information was supplied relating to ethical approvals (i.e., approving body and any reference numbers):

Patient clinical data were collected as part of the quarantine monitoring measures with at the Rafik al Hariri University Hospital (RHUH) with the support of the Lebanese Ministry of Public Health (MoPH). Required written informed consent was waived due to the pressing need for data collection in the early stages of the outbreak. This study was approved by the Institutional Review Board (IRB) of the Lebanese American University IRB#LAU.SAS.ST2.19/May/2020.

The following information was supplied regarding the deposition of DNA sequences:

Data is available at GSAID: EPI_ISL_450508, EPI_ISL_450509, EPI_ISL_450510, EPI_ISL_450511, EPI_ISL_450512, EPI_ISL_450513, EPI_ISL_454420, EPI_ISL_450514, EPI_ISL_450515, EPI_ISL_450516, EPI_ISL_450517.

Accessing the GISAID databases requires users to fill out a registration form and accepting GISAID’s terms of use and database access agreement. Upon filing the request and subsequent review, GISAID will provide the user with unique credentials in an activation email, enabling access to the EpiCoV and EpiFlu databases and all the data included. All genomic data are identifiable based on a unique accession ID known as the EPI accession numbers.

The following information was supplied regarding data availability:

Data is available at GSAID: EPI_ISL_450508, EPI_ISL_450509, EPI_ISL_450510, EPI_ISL_450511, EPI_ISL_450512, EPI_ISL_450513, EPI_ISL_454420, EPI_ISL_450514, EPI_ISL_450515, EPI_ISL_450516, EPI_ISL_450517.

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
