# Peer review of "Genomic characterization and phylogenetic analysis of the first SARS-CoV-2 variants introduced in Lebanon"

_PeerJ, doi:10.7717/peerj.11015_

## Round 0.1 · original submission · Minor Revisions

Please, pay attention to all suggestions and comments of all three reviewers. I would like to get your response to the criticisms of the first reviewer, especially his request to justify your phylogenetic methods.

·

Basic reporting

L28 in the abstract "Presently, and reached almost 15 million cases." This sentence is incomplete and does not make sense.
L30 "it has so far managed the initial SARS-CoV-2 wave", this line needs to be restructured to "it has so far managed to cope with the initial SARS-Cov-2 wave".

Experimental design

The study's main result is a phylogenetic analysis of the strains recovered from isolates obtained from Lebanese patients. There is no mention in the methods section how the short sequenced reads are assembled into whole viral genome sequences to infer a phylogeny. The only detail given was the use of BWA MEM aligner which only aligns sequenced reads to a reference.

What are the justifications for using IQTree as a phylogeny inference? have the authors tried also bayesian methods (such as MrBayes?)? There are many studies benchmarking tree construction software which often show superior quality of RAxML over IQTree.

If the main results of this study are to be based on the inferred phylogeny it is to my opinion some comparisons between different inference tools to be performed.

The long branches in the reported tree is a bit concerning, have the authors tried using different reference viral genomes to map the different strains to?

Was there any particular reason to use SNP-sites? did the authors try using GATK-3?

The methods are missing details on how variants were called to be SNPs, what thresholds were used? and what was done for quality control other than filtering out low quality reads? was sequencing depth taken into account?

Validity of the findings

The findings discussed were consistent with the demonstrated results.

Additional comments

Overall the study offers some insights on how the the novel coronavirus was introduced and spread in Lebanon, the manuscript should warrant publication upon revisions.

Reviewer 2 ·

Basic reporting

Line 40: No comma is needed after “which”

Line 53: “myaglia” needs to be written as “myalgia”

Line 66: Lebanese needs to change to Lebanese individuals

Line 74: The date range might be better if written as 21 Feb 2020- 15 March 2020

Line 75: In “with at”, “at” is redundant

Line 197: Before: “China CDC”; After: “Chinese Center for Disease Control and Prevention (China CDC):

Line 201: “the” in “or the in” is redundant

Line 231: Before: “predominate”; After: “predominant“

Line 240: Before: “entry 614G”; After: “entry of the 614G”

Experimental design

Before: “are also reflective of the situation in regions that lack genomic and epidemiological data, such as Iran.”

After: “are also reflective of the situation in regions that lack genomic and epidemiological data.”

Reason: There are already several studies by Iranian researchers with sequenced patient data.

What are the other applications of this work? If there's other applications, please add. For example, as a preventative measure, for vaccine design, or to show which countries were the first to have the outbreak (e.g. based on fewer mutations).

Validity of the findings

Line 213-220: It would be helpful to add information about the samples, and recent travel history. For instance, Wuhan 412981 (EPI_ISL_412981) was sequenced in Jan 18 2020 (no recent travel to Iran), EPI_ISL_408482 in Jan 19 2020 (no recent travel to Iran), and EPI_ISL_412965 in Feb 2 2020 (recent travel to Iran), in Figure 2, or as a separate table or figure. Also, an unrooted representation of the tree may show the relatedness of the nodes better.

It might be helpful to add Jalview alignments showing some or all of SNVs.

·

Basic reporting

There are several places throughout the text with small language typos that may make it difficult for an international audience to understand. Some examples where the language could be improved include lines 29 and 31.

Experimental design

The work presented by Feghali et al in “Genomic characterization and phylogenetic analysis of the first SARS-CoV-2 variants introduced in Lebanon” is important, timely, and likely to be of interest to a wide audience. It addresses an important knowledge gap as there do not appear to be any similar studies investigating SARS-CoV-2 the area. The approaches are well suited to investigate the source of SARS-CoV-2 in Lebanon and the conclusions (that there were likely multiple introductions) are well-supported by the results. They were very thorough. The weaknesses of the manuscript, if any, can be resolved with some additional clarification and elaboration as outlined below.

Line 81- Do you mean qRT-PCR?
Line 84- I was under the impression that RdRp stood for RNA dependent RNA polymerase. Please confirm.
Lines 87- While the volume is helpful, it would be more meaningful to provide the concentration (if known). For example, 5ul of RNA extract could be 20ng of RNA or 200ng. Were the primers 5mM or 100mM?
Line 104- How was the consensus generated? Is that a feature in BWA-MEM?
Line 107- Can you please provide more information about how the 50 reference sequences were chosen. Did you choose the 50 most closely related sequences?
Line 109- Please include a table of metadata for all 50 reference sequences included in this study. The following data should be included if available: name, accession number, date of collection, location of origin, travel history.
Line 192- Do you mean M33F?
Line 213 and 231- The results of the Pangolin analysis are used to draw significant conclusions. Can you show those results? Even if you just add the lineage/clade to one of the other tables.

Validity of the findings

The conclusions of this paper are well supported by the results. However, there are a few places where additional elaboration is needed.

Lines 152-158- According to your figure, the bootstrap support for the clustering pattern seems to be well-supported. Can you please specify the values in the text or on the figure? Also, please reference figure 2 here.
Line 165- I am not sure what you mean by “S1 exhibited less genomic diversity than S2 and S8 when compared to each other”. Do you mean that S1 was less divergent from the other sequences that it clustered with?
Line 210- Why do you include that there are 2 main lineages if your sequences fall into 3 clades?

---

## Round 0.2 · accepted · Accept

My personal congratulations and apologies for so long process of reviewing.

·

Basic reporting

looks good!

Experimental design

looks good!

Validity of the findings

looks good!

Additional comments

the manuscript looks much refined after addressing reviewers comments.

Reviewer 2 ·

Basic reporting

For supplementary figure 1, I suggest using a better quality format if possible. You can see the higher quality formats from here:
http://www.jalview.org/help/html/io/export.html

Experimental design

My previous comments have been addressed satisfactorily.

Validity of the findings

My previous comments have been addressed.